# Susceptibility profile of *Aeromonas* spp. from clinical strains isolated in French Guiana

Vincent Sainte-Rose,[1] Alexis Daude,[1] Tojoniaina H. Andriamandimbisoa,[1] Romain Blaizot,[2,3] Stéphanie Houcke,[4] Olivier Lesens,[5] Jean de la Croix Jaonasoa,[1] Daniel Selenge Kaozi,[1] Karamba Sylla,[5] Magalie Demar,[1,3] Hatem Kallel[3,4]

**ABSTRACT**  *Aeromonas* spp. are gram-negative bacilli of the Gammaproteobacteria class and Aeromonadaceae family. These bacteria are a public health challenge, particularly in South America. They harbor chromosomal β-lactamases belonging to classes B, C, and D, whose presence varies among species. This retrospective study aimed to evaluate the minimum inhibitory concentrations (MICs) of piperacillin-tazobactam (PTZ) and cefotaxime (CTX) for clinical strains of *Aeromonas* spp. A secondary objective was to assess the resistance profile of these strains to cefepime (FEP), aztreonam (AZT), ciprofloxacin (CIP), levofloxacin (LEV), and trimethoprim-sulfamethoxazole (SXT). The study was conducted at the Microbiology Laboratory of Cayenne Hospital (French Guiana) from January 2020 to December 2023. All isolated clinical *Aeromonas* strains, regardless of species or sample type, were included. In total, 99 strains were analyzed, comprising *Aeromonas hydrophila* (*n* = 79), *Aeromonas caviae* (*n* = 10), *Aeromonas veronii* (*n* = 6), and other *Aeromonas* spp. (*n* = 4). The median PTZ MIC was 0.5 mg/L for *Aeromonas* spp. overall, with six strains (6%) exhibiting MICs of >8 mg/L. For *A. hydrophila*, the median PTZ MIC was also 0.5 mg/L, lower than the median reported by the European Committee on Antimicrobial Susceptibility Testing (EUCAST) for strains of 4 mg/L. For CTX, the median MIC for all *Aeromonas* spp. was 0.064 mg/L, with four strains (4%) having MICs of >4 mg/L. The median ceftazidime (CAZ) MIC was 0.125 mg/L. No resistance to AZT was detected, and 3% high-dose susceptibility to FEP was found. Resistance rates to other antibiotics were low: 6% to CIP, 6% to LEV, and 6% to SXT. In conclusion, our study demonstrated low resistance rates in *Aeromonas* spp. to PTZ (6%), CTX (4%), and CAZ (4%). These findings suggest that PTZ and CTX could be effective against *Aeromonas* infections, despite the lack of established clinical breakpoints by EUCAST.

**IMPORTANCE**  *Aeromonas* spp. are clinically significant, particularly in tropical climates. This study provides essential MIC distribution data for piperacillin-tazobactam and cefotaxime, offering valuable insights for guiding empirical treatment choices. Understanding these susceptibility patterns is crucial for optimizing antimicrobial therapy, particularly in regions with limited resistance surveillance and standardized guidelines. By analyzing a large collection of clinical isolates, our findings contribute to evidence-based decision-making for *Aeromonas* infections and emphasize the importance of continued surveillance of antimicrobial susceptibility. These findings suggest that PTZ and CTX may be therapeutic options for *Aeromonas* infections, supporting their potential use in clinical practice.

**KEYWORDS**  *Aeromonas* spp., *A. hydrophila*, cefotaxime, piperacillin-tazobactam, MIC

**Peer Reviewer** Ayesha Khan, UCI Health, Orange, California, USA

Address correspondence to Vincent Sainte-Rose, vincent.sainterose@ch-cayenne.fr.

Magalie Demar and Hatem Kallel contributed equally to this article.

The authors declare no conflict of interest.

*A*eromonas spp. are gram-negative bacilli that belong to the Gammaproteobacteria class and the Aeromonadaceae family. This genus comprises 36 species, of which

*Aeromonas hydrophila* is the most well-known (1). They are ubiquitous microorganisms commonly found in aquatic environments such as freshwater, brackish water, wastewater, soil, and in the microbiota of certain animals (2, 3). This genus can be involved in human pathology, especially in cutaneous, intestinal, pulmonary, and bloodstream infections (4, 5). Regarding antibiotic resistance, *Aeromonas* spp. possess three chromosomal β-lactamases belonging to classes B, C, and D, whose presence varies across different species. In *A. hydrophila*, all three enzymes are present at very low levels, conferring resistance to amoxicillin ± clavulanic acid, ticarcillin, and cefalotin and reduced susceptibility to imipenem in wild strains (6, 7). A higher level of resistance may be observed due to the hyperproduction of chromosomal cephalosporinase (7). In other common species, *A. veronii* possesses class B and D β-lactamases, and *Aeromonas caviae* contains class C and D enzymes. In addition, resistance to colistin is increasing in relation to the presence of the mcr-1, mcr-3, and mcr-5 variants (8).

Globally, *Aeromonas* spp. are susceptible to piperacillin (PIP) ± tazobactam (piperacillin-tazobactam [PTZ]), cefotaxime (CTX), ceftriaxone (CTR), ceftazidime (CAZ), ciprofloxacin (CIP), amikacin (AMK), gentamicin, and trimethoprim-sulfamethoxazole (SXT) (5, 9, 10).

At the Cayenne Hospital in French Guiana (South America), *Aeromonas* spp. are most frequently isolated from polymicrobial infections of wounds and soft tissues, often related to traffic accidents or snake bites (11, 12), requiring broad-spectrum treatment such as PTZ, CTX, or CAZ. However, the European Committee on Antimicrobial Susceptibility Testing (EUCAST) did not provide clinical minimum inhibitory concentration (MIC) breakpoints to PTZ and CTX (13). Whereas, in clinical practice, we rely on their pharmacokinetic/pharmacodynamic (PK/PD) breakpoints to assess susceptibility.

We conducted this study to investigate the susceptibility of *Aeromonas* to these antibiotics. The primary objective was to assess the MICs of clinical strains of *Aeromonas* spp. for PTZ and CTX and to compare them to the CAZ MIC (for which EUCAST already defined a clinical breakpoint). The secondary objective was to consider the resistance profile of clinical strains of *Aeromonas* spp. to cefepime (FEP), aztreonam (AZT), CIP, levofloxacin (LEV), and SXT. These antibiotics are systematically tested and can be used as oral relays (SXT, LEV, and CIP).

## MATERIALS AND METHODS

### Bacterial strain

Our retrospective study was conducted at the Microbiology Laboratory of the Cayenne Hospital (French Guiana) from January 2020 to December 2023. We included all isolated clinical *Aeromonas* strains, regardless of species or samples. Overall, out of 177 strains isolated, 99 strains were analyzed. The others were excluded due to lack of conservation. Analyzed strains include *A. hydrophila* (n = 79), *Aeromonas caviae* (n = 10), *Aeromonas veronii* (n = 6), and other *Aeromonas* spp. (n = 4) (Fig. 1). The clinical samples were mucocutaneous (n = 55), biopsy (n = 33), blood culture (n = 8), stool (n = 4), bile (n = 1), pleural fluid (n = 1), eye (n = 1), and abdominal drain (n = 1).

### Identification

Positive cultures were identified by mass spectrometry (Bruker MALDI Biotyper) using the direct transfer method. Fresh colonies were transferred to a polished steel MSP 96 target (Bruker Daltonik) and coated with one 70% formic acid microliter. After drying, 1 µL of saturated α-cyano-4-hydroxycinnamic acid matrix (Bruker Daltonik) was added to the spot.

Spectra were acquired and analyzed using flexcontrol software (Bruker Daltonik). According to the supplier's recommendations, a score of ≥2 was considered reliable for the species; a score between 1.7 and 2.0 was deemed reliable for the genus.

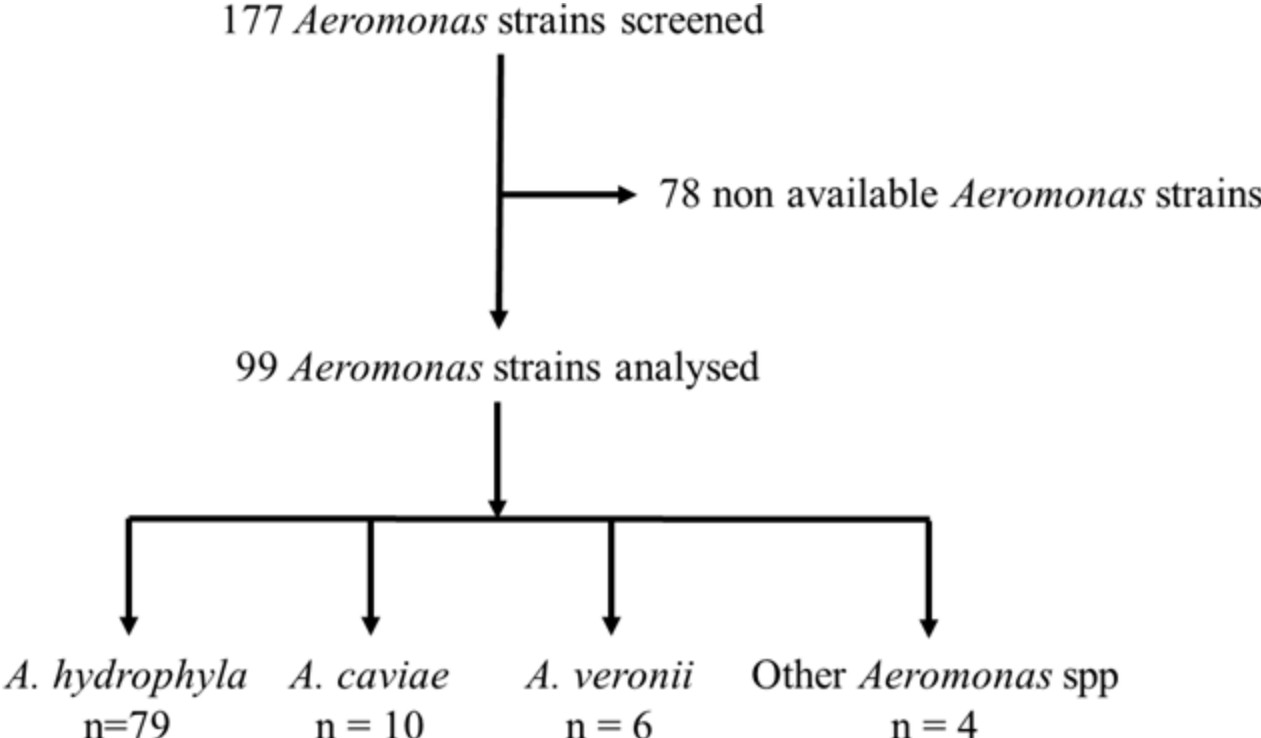

**FIG 1** Flowchart of the investigated *Aeromonas* strains.

## Antibiogram and MIC

Antibiograms were performed using the agar diffusion disk method for FEP, ATM, SXT, CIP, and LEV (Oxoid disk, Thermo Fisher Diagnostics). E-test strips were used for CAZ and CTX (MIC strips, i2a). For both methods (antibiotic disks and MIC strips), we used Mueller-Hinton agar (bioMérieux, Marcy-l'Étoile). A 0.5 McFarland inoculum was prepared, followed by incubation at 35°C ± 2°C for 16–24 h in an aerobic atmosphere. Inhibition diameter measurements were automated using the ADAGIO system (Bio-Rad).

Piperacillin/tazobactam MICs were determined using microdilution strips (UMIC, Biocentric/Bruker). Incubation was performed at 35°C ± 2°C for 16–20 h.

We used EUCAST 2018 (for strains isolated in 2020) and EUCAST 2020 (for strains isolated in 2021–2023) to interpret susceptibility (Table 1). No changes are observed between the 2018 and 2020 versions regarding *Aeromonas* spp. susceptibility breakpoints.

We also compared PTZ MICs with those measured by the EUCAST for *A. hydrophila* (n = 55).

## RESULTS

### Piperacillin/tazobactam MIC

Overall, the median PTZ MIC for all combined *Aeromonas* spp. was 0.5 mg/L (Fig. 2A). Six strains (6%) exhibited MICs of >8 mg/L, including three *A. hydrophila* strains (numbers 39, 63, and 116), two *A. caviae* strains (numbers 111 and 114), and one *A. veronii* strain (number 83) (Table 2). The three *A. hydrophila* strains showed high MICs to CTX and CAZ and decreased susceptibility to FEP. However, they were susceptible to AZT, fluoroquinolones, and SXT. In the remaining three species (two *A. caviae* spp. and one *A. veronii* sp.), no co-resistance with any of the tested antibiotics was detected.

For *A. hydrophila*, the median PTZ MIC was 0.5 mg/L (n = 79), which was lower than the median MIC reported for strains analyzed by EUCAST (4 mg/L, n = 55) (Fig. 2B).

**TABLE 1** Summary of breakpoints according to EUCAST

| Antibiotics | Methodology | MIC breakpoints (mg/L) | | Zone diameter breakpoints (mm) | |
|---|---|---|---|---|---|
| | | S ≤ | R > | S ≤ | R > |
| Cefepime | Clinical breakpoint | 1 | 4 | 27 | 24 |
| Ceftazidime | Clinical breakpoint | 1 | 4 | 24 | 21 |
| Aztreonam | Clinical breakpoint | 1 | 4 | 29 | 26 |
| Ciprofloxacin | Clinical breakpoint | 0.25 | 0.5 | 27 | 24 |
| Levofloxacin | Clinical breakpoint | 0.5 | 1 | 27 | 24 |
| Trimethoprim-sulfamethoxazole | Clinical breakpoint | 2 | 4 | 19 | 16 |
| Piperacillin-tazobactam | PK/PD cut-off values | 8 | 16 | –[a] | – |
| Cefotaxime | PK/PD cut-off values | 0.5 | 0.5 | – | – |

[a]"–" indicates that no diameter was tested for these molecules.

## Cefotaxime MIC

The median CTX MIC (all species included) was 0.064 mg/L (Fig. 3A). Four strains (4%) (Table 2) had MICs of >0.5 mg/L (corresponding to the threshold of susceptibility to CTX based on clinical PK/PD concentrations by EUCAST). These were three *A. hydrophila* strains (strains 39, 63, and 116) and one *A. caviae* strain (strain 85). The latter has a borderline MIC to PTZ (8 mg/L) and decreased susceptibility to all the other tested molecules.

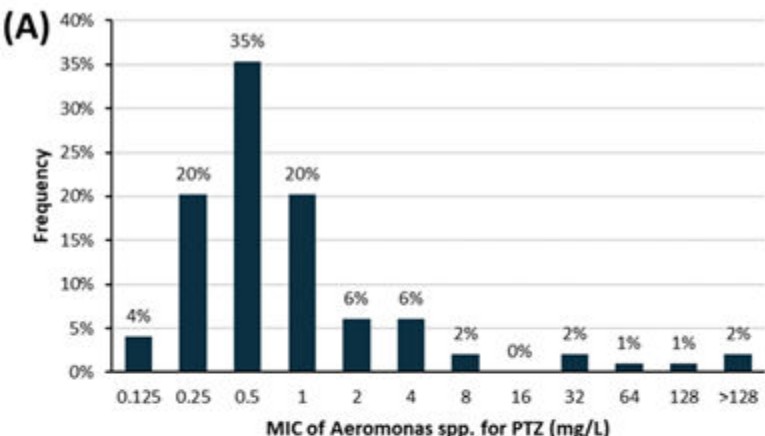

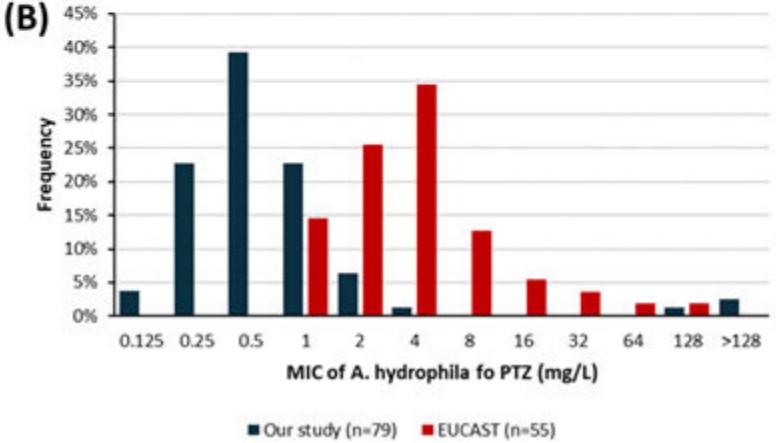

**FIG 2** (A) The MIC distribution for piperacillin-tazobactam for all *Aeromonas* spp. (*n* = 99). (B) MIC distribution for the genus *A. hydrophila* (*n* = 79).

**TABLE 2** Summary of strains presenting resistance according to EUCAST breakpoints[a,b,c]

| Number | Ident. | MIC PIT | MIC CTX | MIC CAZ | FEP | ATM | CIP | LEV | SXT |
|---|---|---|---|---|---|---|---|---|---|
| 3 | *A. veronii* | 4.00 | 0.03 | 0.13 | S | S | **R** | **R** | **S** |
| 19 | *A. hydrophila* | 1.00 | 0.09 | 0.19 | S | S | **R** | **R** | **R** |
| 54 | *A. hydrophila* | 0.50 | 0.19 | 0.13 | S | S | **R** | **SHD** | **R** |
| 72 | *A. hydrophila* | 0.50 | 0.19 | 0.13 | S | S | **R** | **R** | **R** |
| 39 | *A. hydrophila* | **256.00** | **64.00** | **512.00** | **SHD** | S | S | S | S |
| 63 | *A. hydrophila* | **128.00** | **32.00** | **8.00** | **SHD** | S | S | –[d] | S |
| 83 | *A. veronii* | **32.00** | 0.05 | 0.25 | S | S | S | S | S |
| 85 | *A. caviae* | 8.00 | **8.00** | **12.00** | S | S | **R** | **R** | **R** |
| 111 | *A. caviae* | **32.00** | 0.06 | 0.25 | S | S | S | S | S |
| 114 | *A. caviae* | **64.00** | 0.05 | 0.38 | S | S | S | S | S |
| 116 | *A. hydrophila* | **256.00** | **16.00** | **24.00** | **SHD** | S | S | S | S |

[a]PK/PD breakpoints of Comité de l'Antibiogramme de la Société Française de Microbiologie/EUCAST forpiperacil-line-tazobactam (S ≤8 mg/L – R >16 mg/L) and cefotaxime (S ≤1 mg/L – R >2 mg/L) and clinical breakpoints for ceftazidime (S ≤1 mg/L – R >4 mg/L).
[b]R, resistant; S, susceptible; SHD, susceptible at high doses.
[c]The boldfaced items indicate that there is a higher level of resistance for these molecules.
[d]"–" indicates that no diameter was tested for these molecules.

The median CTX MIC according to the studied species was 0.064 mg/L ($n$ = 79) for *A. hydrophila* (Fig. 3B).

## Ceftazidime MIC

For all combined species, the median CAZ MIC was 0.125 mg/L (Fig. 4A). Four strains (4%) (Table 2) had MICs of >4 mg/L (corresponding to the threshold for CAZ susceptibility based on clinical concentrations of the *Aeromonas* spp. genus proposed by EUCAST). These are the same strains with high MICs to cefotaxime.

The median CAZ MIC was 0.125 mg/L for *A. hydrophila* ($n$ = 79) (Fig. 4B).

## Other antibiotics

No resistance was detected for AZT (Fig. 5). Three out of 97 (3%) were susceptible at high doses to FEP (Fig. 5). Six of 98 (6%) strains were resistant to CIP; 5 of 89 (6%) strains were resistant to LEV; and 5 of 90 (6%) were resistant to SXT (Fig. 5).

Overall, associated resistance to CIP, LEV, and SXT was observed in five strains: four *A. hydrophila* strains and one *A. caviae* strain (Table 2). The four *A. hydrophila* strains showed no decreased susceptibility to β-lactams.

## DISCUSSION

This study showed low MICs for PTZ, CTX, and CAZ in *Aeromonas* spp. with median MIC values of 0.5, 0.064, and 0.125 mg/L, respectively. However, eight strains (8 of 99) showed high MICs against certain β-lactam antibiotics (Table 2). These include three *A. hydrophila* strains (3 of 79, 4%) (strains 39, 63, and 116). These strains showed decreased susceptibility to all tested β-lactams except AZT, while FEP was categorized as "susceptible at high doses" according to EUCAST breakpoint. This phenotype may correspond to hyperproduction of the chromosomal cephalosporinase CepH, likely driven by a regulatory mutation. This phenomenon has been previously described, notably following the use of CTX (14, 15).

The same resistance phenotype was also observed in one *A. caviae* strain (1 of 10) (strain 85), likely related to the derepression of its chromosomal cephalosporinase, CAV-1 (16). The limited number of *A. caviae* strains in this study ($n$ = 10) makes it difficult to accurately estimate the resistance level in this species.

The last or remaining three strains (three out of six) showed a high MIC only with PTZ. These included one *A. veronii* strain (strain 83) and two *A. caviae* strains (strains 111 and 114). This resistance profile was previously associated with the class D β-lactamase (oxacillinase) in *Aeromonas sobria* (17). Since *A. veronii* and *A. caviae* also produce

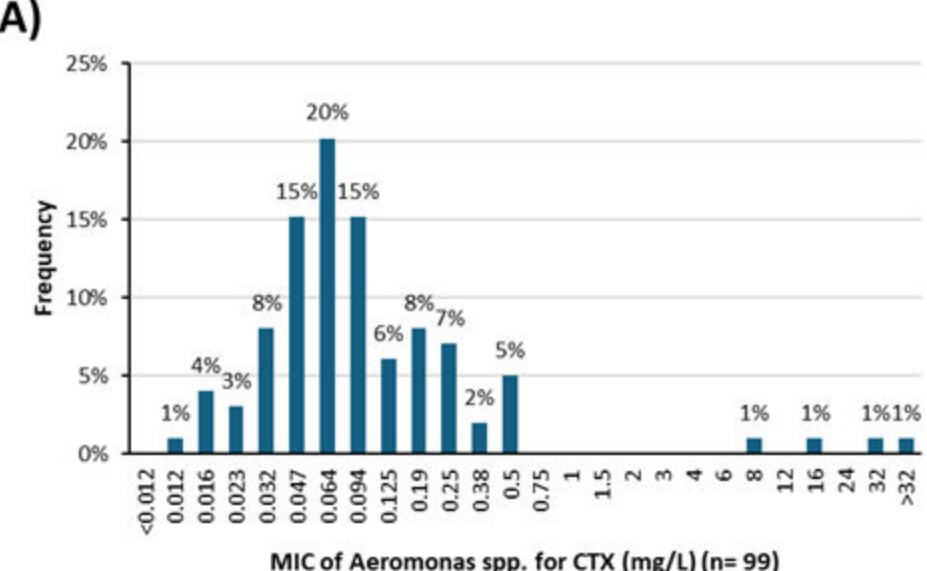

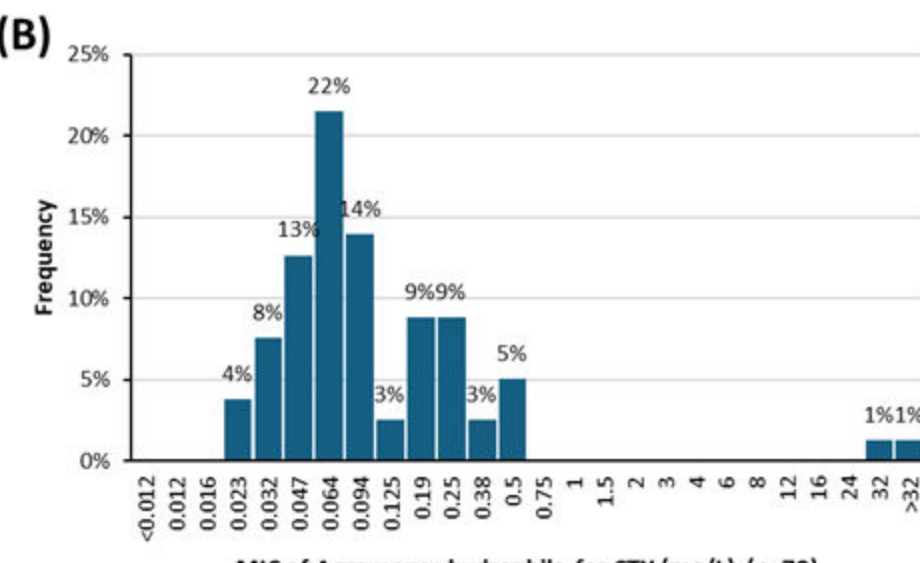

FIG 3  (A) The MICs distribution for cefotaxime (CTX) for all *Aeromonas* spp (*n* = 99). (B) MICs distribution according to *Aeromonas hydrophila* (*n* = 79).

oxacillinase, it is hypothesized that a mutation leading to overproduction of this enzyme might specifically affect PTZ without impacting cephalosporins. Further studies are needed to confirm this hypothesis. The small number of *A. veronii* strains (*n* = 6) also makes it challenging to accurately assess the resistance level in this species.

In our study, the PTZ MICs appear to be slightly lower than those reported for European strains tested by EUCAST (Fig. 2). For *A. hydrophila*, the median PTZ MIC was 0.5 mg/L (72 strains) compared to the median of 4 mg/L reported by EUCAST (55 strains). A study by Lamy et al. (5) conducted in mainland France showed that 12% of 25 *A. hydrophila* strains were resistant to PTZ, while only 4% of our strains were resistant. A study by Castelo-Branco et al. (18) in the northeastern region of Brazil reported PTZ MICs for *A. hydrophila* that closely matched our findings. In fact, 18 out of 19 clinical strains exhibited PTZ MICs ranging from 0.06 to 4.0 mg/L. This supports the observation that *A. hydrophila* has low PTZ MICs in our region, indicating that PTZ can be safely employed

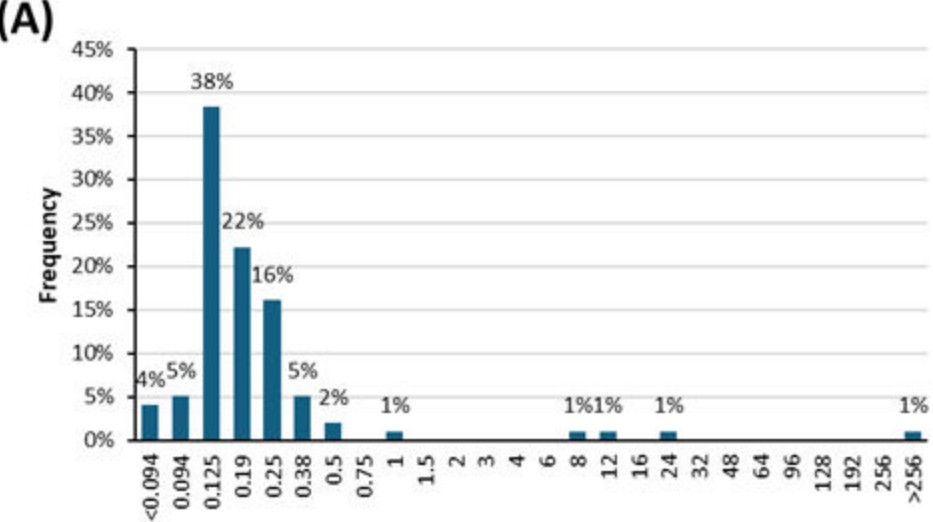

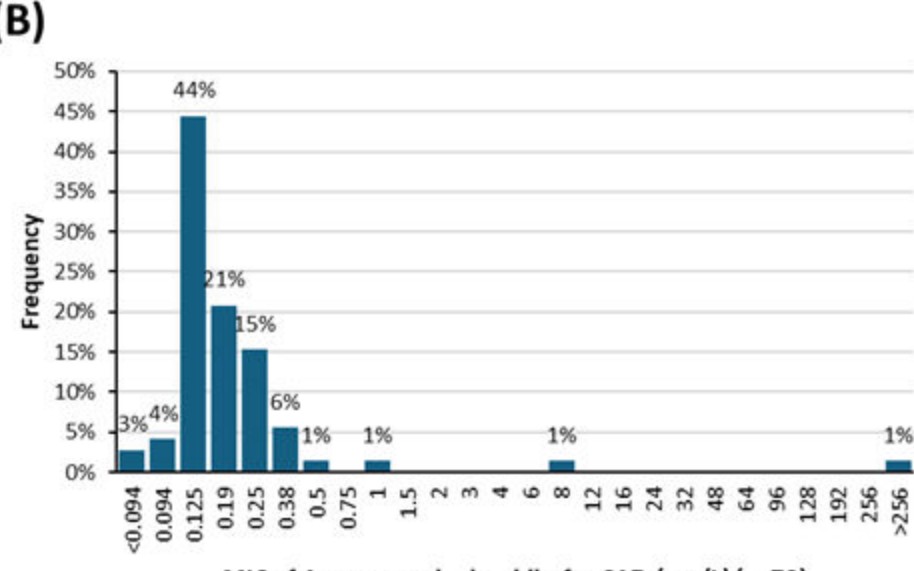

**FIG 4** (A) The MIC distribution for ceftazidime (CAZ) for all *Aeromonas* spp. (*n* = 99). (B) MIC distribution according to *Aeromonas hydrophila* (*n* = 79).

in this context. Resistance to LEV, CIP, and SXT was approximately 6% across all species combined. Lamy et al. (5) showed 13% resistance to cotrimoxazole and 8% resistance to LEV in mainland France.

As often observed, fluoroquinolone resistance may result from mutations in the *gyrA* and *parC* genes or the transmission of the *qnr* gene on cassettes carried by a class 1 integron (1, 19). This class 1 integron may also harbor a *dhfr* gene which confers resistance to cotrimoxazole. Such a scenario could apply to our quinolone-resistant strains, where four out of five strains were also resistant to cotrimoxazole.

*Aeromonas* infections are frequently described in France primarily linked to skin and soft tissue infections that could result from road accidents or snake bites (12). Houcke et al. (11) reported 172 patients hospitalized due to snake bites between 2016 and 2021, and 37.5% of these patients presented an infection caused by *A. hydrophila*. Furthermore, the recommended antibiotic therapy in this setting for necrotizing fasciitis or septic shock consists of a combination of first-line PTZ, CTX, FEP, or CAZ along with AMK. As

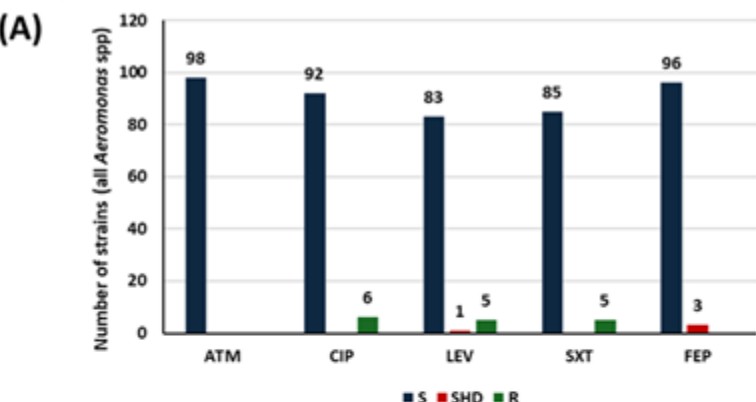

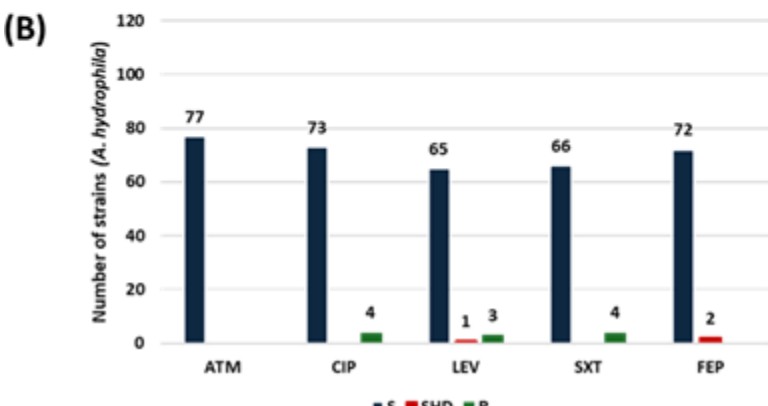

**FIG 5** Antibiotic susceptibility of the tested *Aeromonas* strains (A) and *A. hydrophila* (B), according to EUCAST guidelines. ATM, aztreonam; CIP, ciprofloxacin; FEP, cefepime; LEV, levofloxacin; R, resistant; S, susceptible; SHD, susceptible at high doses; SXT, trimethoprim-sulfamethoxazole.

PTZ and CTX have no established clinical breakpoints in the EUCAST guidelines, it was therefore important to evaluate their MICs and resistance profiles.

Our data showed that PTZ has low MICs against *Aeromonas* spp. and a low resistance rate (4%) based on the clinical PK/PD concentrations recommended by EUCAST. These data support the use of PTZ as an effective treatment for *A. hydrophila* infections.

The same observations can be applied to CTX, which also has a 4% resistance rate and has overall MICs lower than those of PTZ and CAZ. Therefore, CTX can be considered a relevant treatment option against *Aeromonas* infections. However, caution is advised as some studies do not recommend its use as monotherapy in severe infections with *A. hydrophila* and *A. caviae* due to their natural cephalosporinases (15, 20).

Furthermore, it is important to note that in our retrospective study, several methods were used to determine antibiotic susceptibility, including disk diffusion, E-test strips, and microdilution strips. As disk diffusion is not the reference method, further studies are needed to confirm our results regarding the following antibiotics: ATM, FEP, CIP, LEV, and SXT.

## Conclusion

Our study showed low resistance rates in *Aeromonas* spp. for PTZ (6%), CTX (4%), and CAZ (4%). Therefore, despite the absence of EUCAST clinical breakpoints for PTZ and CTX, these agents could be used for treating infections caused by *Aeromonas* spp. We also observed low resistance rates to other antibiotics that can be used as oral alternatives, including LEV (6%), CIP (6%), and SXT (6%). Lastly, no resistance to ATM was found, and 3% of the strains were susceptible to high doses of FEP.

## AUTHOR AFFILIATIONS

[1]Laboratory of Microbiology, French Guiana University Hospital, Cayenne, French Guiana

[2]Department of Dermatology, French Guiana University Hospital, Cayenne, French Guiana

[3]Tropical Biome and immunopathology CNRS UMR-9017, Inserm U 1019, Université de Guyane, Cayenne, French Guiana

[4]Intensive Care Unit, French Guiana University Hospital, Cayenne, French Guiana

[5]Tropical and Infectious Diseases Department, French Guiana University Hospital, Cayenne, French Guiana

## AUTHOR ORCIDs

Vincent Sainte-Rose  http://orcid.org/0000-0002-1170-0817

## AUTHOR CONTRIBUTIONS

Vincent Sainte-Rose, Conceptualization, Data curation, Formal analysis, Methodology, Resources, Writing – original draft, Writing – review and editing | Alexis Daude, Investigation, Resources | Tojoniaina H. Andriamandimbisoa, Visualization | Romain Blaizot, Validation, Visualization | Stéphanie Houcke, Validation, Visualization | Olivier Lesens, Validation, Visualization | Jean de la Croix Jaonasoa, Visualization | Daniel Selenge Kaozi, Visualization | Karamba Sylla, Validation, Visualization | Magalie Demar, Conceptualization, Supervision, Validation, Writing – review and editing | Hatem Kallel, Conceptualization, Supervision, Validation, Writing – review and editing

## ADDITIONAL FILES

The following material is available online.

### Open Peer Review

**PEER REVIEW HISTORY (review-history.pdf).** An accounting of the reviewer comments and feedback.

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
