## [Reviewer comments · Microbiology Spectrum]

Microbiology Spectrum

Susceptibility Profile of *Aeromonas* spp. From Clinical Strains Isolated in French Guiana

Vincent SAINTE-ROSE, Alexis DAUDE, Tojoniaina ANDRIAMANDIMBISOA, Romain Blaizot, Stéphanie Houcke, Olivier Lesens, Jean de la Croix Jaonaso, Daniel Selenge Kaozi, Karamba Sylla, Magalie Demar, and Hatem Kallel

Corresponding Author(s): Vincent SAINTE-ROSE, University Hospital, French Guiana

Review Timeline:

Submission Date:	February 17, 2025
Editorial Decision:	March 13, 2025
Revision Received:	May 6, 2025
Editorial Decision:	June 17, 2025
Revision Received:	July 15, 2025
Accepted:	July 31, 2025

Editor: Florence Doucet-Populaire

Reviewer(s): Disclosure of reviewer identity is with reference to reviewer comments included in decision letter(s). The following individuals involved in review of your submission have agreed to reveal their identity: Ayesha Khan (Reviewer #1)

Transaction Report:

DOI: <https://doi.org/10.1128/spectrum.00350-25>

Re: Spectrum00350-25 (Susceptibility Profile of Aeromonas spp. From Clinical Strains Isolated in French Guiana)

Dear Dr. Vincent SAINTE-ROSE:

Thank you for the privilege of reviewing your work. Below you will find my comments, instructions from the Spectrum editorial office, and the reviewer comments.

Revision Guidelines

Sincerely,
Florence Doucet-Populaire
Editor
Microbiology Spectrum

Reviewer #1 (Comments for the Author):

I was pleased to review the manuscript entitled "Susceptibility Profile of Aeromonas spp. From Clinical Strains Isolated in French Guiana".

Aeromonas species are increasingly recognized as important pathogens in human infections. There is a paucity of data on their clinical characteristics and antimicrobial susceptibility profiles. Some species are reported to be intrinsically resistant due to

chromosomal beta-lactamases (e.g. metallo-beta-lactamase, AmpC, or OXA enzymes). However, the prevalence of these intrinsic resistance mechanisms across species is unclear. There is also insufficient data on the correlation of resistance genotype and phenotype. This study evaluated antimicrobial susceptibility of n=99 diverse *Aeromonas* strains collected at a hospital in French Guiana between 2020-2023. A robust characterization of antimicrobial susceptibility patterns of contemporary clinical *Aeromonas* strains across different species is important and clinically useful.

General major comment on manuscript: The most important feedback for the authors is that there needs to be at least a subset of isolates (if not for all 99 strains) for which ALL 3 AST method data are available: MIC from broth microdilution (BMD), zone diameter from disk diffusion and Etest MIC. Then, the BMD can be used to generate MIC distributions and compare susceptibility profiles to different agents across species. Additionally, the performance of Etest and disk diffusion can be evaluated relative to the reference BMD method. If not for all antimicrobial agents, then at least for the most clinically important agents (cefepime, ceftaz, pip-tazo etc).

Below are comments about further experiments and/or edits the authors can make to improve the overall quality of the study. I believe there are many more opportunities for the authors to increase the clinical utility of this study.

1. Please add a table in the methods that lists all the clinical MIC and disk diffusion BREAKPOINTS that were used in the study for each antimicrobial agent to interpret susceptibility testing results.

It can otherwise be very difficult for readers to follow along. This is also the most important piece to add clinical utility to these in vitro experiments.

2. I would highly recommend using only two types of clinical breakpoints to interpret susceptibility results: the CLSI M45 guidelines and EUCAST. Those are the two most common standards followed by clinical laboratories around the world. It keeps things simple and broadly applicable. I think the various breakpoints from CA-SFM are confusing and not widely used outside of France.

In Table 1, make sure to include a column to specify if the MIC or disk diffusion breakpoints are from CLSI or EUCAST. It would be informative to have this table contrasting the CLSI versus EUCAST breakpoints for the agents where they are available. If there are differences between CLSI and EUCAST breakpoints for the same agent, then perform analysis with both S/I/R or S/R/ATU criteria to compare how many strains would categorize as susceptible or non-susceptible.

3. It can be difficult to compare antimicrobial susceptibility profiles across drugs because 3 different methods were used inconsistently. Some drugs were tested with disk diffusion (CAZ, FEP, ATM, SXT, CIP, LEV) while others were tested with Etest (CTX), some had both disk and Etest (CAZ, CTX, FEP) and TZP was tested with broth microdilution. Given that the goal of the study is to characterize the susceptibility profiles of these strains and compare susceptibility to different antimicrobials, it is important to use standard methods with appropriate quality control strains.

- Is it possible to test all antimicrobial agents (or the most important ones used clinically) with Broth Microdilution (BMD)? BMD is considered the gold standard reference method for susceptibility testing.

Considering that this is a single center study with microbiological data (in the absence of clinical data), it is important to make sure the right methods were used for AST to be able to compare susceptibility across agents and species. If resources are limited, choose 25-30 representative isolates of *Aeromonas hydrophila* based on the MIC/ disk diffusion data already available (across a range of MICs or zone diameters, some more susceptible, some resistant) plus all 10 *A. caviae* strains, 6 *A. veronii* strains and the 4 other *Aeromonas* spp strains. Even if testing is performed on a smaller subset of strains (~40-50 strains) rather than all 99 strains, the conclusions drawn will be more robust with reliable data.

- Either with all 99 strains or with a smaller representative subset of strains, I think it is important to have data to compare all three AST methods for the most clinically important antimicrobials. BMD, Etest, and disk diffusion. If resources are limited, can prioritize cefepime, piperacillin-tazobactam, ceftazidime, bactrim (SXT), aztreonam. If more supplies available, can pick a fluoroquinolone (cipro or levo).

 Use MICs by broth microdilution (BMD) to generate main antimicrobial susceptibility profiling data. Please include the following data points: range of MIC values, MIC50 value, MIC90 value, % susceptible / % intermediate / % resistant for each agent across both CLSI breakpoints and EUCAST (whichever are available). Consider adding a TABLE summarizing the AST data and breaking it down for all strains tested, then across species to compare.

 Evaluate performance of Etest relative to BMD. Compare MIC range and MIC50/MIC90 values between Etest versus BMD. Calculate % essential agreement, % categorical agreement (across both CLSI and EUCAST breakpoints for comparison), very major errors, major errors and minor errors. Can use CLSI M23 criteria for error rates to determine if performance of Etest is acceptable or not. Please consider creating a table summarizing the method evaluation data across antimicrobials and species.

 Evaluate performance of disk diffusion relative to BMD. Calculate % categorical agreement (across both CLSI and EUCAST breakpoints for comparison), very major errors, major errors and minor errors. Create a Scattergram figure to visually show disk-diffusion-to-MIC correlation where the X axis has zone diameters and the Y axis has MIC values. If needed, look at this paper for reference on what the scattergrams can look like for each drug: <https://journals.asm.org/doi/10.1128/jcm.01757-19> (pubmed)

PMID:31996445). Same as before, you can use CLSI M23 criteria to then determine if performance of disk diffusion is acceptable or not.

4. Were quality control strains set up with each testing run? It is important to ensure the potency of the antimicrobials is at expected levels. Consider setting up QC strains with defined MIC ranges for each agent for each day of experiments. For example, consider setting up disk diffusion with ATCC strains like *E. coli* 25922 (CLSI M100 guidelines have tables with MIC and disk diffusion QC ranges for specific ATCC strains) and others that might be needed for each antimicrobial. Read QC results alongside the test strains to make sure data is reliable and MICs aren't falsely low or high.

5. Figure 2 legend says these are global MIC distributions for TZP but I don't believe that is true? Modify accordingly. The word "global" generally means the strains were collected from different geographic regions around the world. Please include # of strains that the graphs are based on in each panel (A-D). For example, how many strains are the distributions in Figure 2A based on? If 99 then specify.

6. What media manufacturers of Mueller Hinton Agar were used for disk diffusion and Etest? Please specify this in the methods section. There can be variability to AST results across different brands of Mueller Hinton Agar. If possible, you can chose a smaller representative subset of isolates (can be as little as 20 strains) to perform disk diffusion and Etest across two or three brands of Mueller Hinton agar. It can show you if these AST methods are precise and reproducible for *Aeromonas*.

7. While comparing performance of disk diffusion or Etest relative to BMD, if there are very major and major errors, consider performing repeat testing to determine if those errors persist or resolve.

8. Consider modifying the MIC distribution graphs after final experiments and include bars for MIC values from BMD and distinct colored bars for MIC values from Etest for comparison. Visually, it would help to see if the Etest tends to undercall or overcall resistance compared to BMD.

9. For all other drugs besides TZP like CTX, CAZ, FEP and ATM-- I think MIC values from broth microdilution are important for more accurate conclusions about the susceptibility profiles of different *Aeromonas* species. Since this is a big piece of manuscript, it seems important to make sure the data is robust. This applies to Figures 3, 4 and 5. Since there is Etest MIC data available also, the graphs can include bars for MICs from BMD versus different color bars for MICs from Etest for comparison.

10. Figure 5: A graph like this is drawing significant conclusions about *Aeromonas* susceptibility to different agents and would need to be based on broth microdilution based MIC data. In this case, there are different methods used for different drugs and the clinical breakpoints are also different so it is very confusing.

Reviewer #2 (Comments for the Author):

Summary of the manuscript: An in vitro MIC survey of 99 clinical isolates of *Aeromonas* spp from French Guiana area.

Major comments: This study is well-designed with agents that have of clinical treatment importance. The data visualization is well organized. However, with the isolates of *A. caviae* of n=10, *A. veronii* of n=6, and other *Aeromonas* spp of n=4, the percent of susceptibility is not supported. Authors may consider present the MIC distributions associated with the 79 *A. hydrophila* isolates only for this report, other species can remain as brief descriptive of mentions, but not for species-specific antibiograms.

Minor comments:

1. Page 2, line 56: CASFM should have a full spell before the abbreviation.

2. Page 5, line 111: ..."oral relays for some CASES..."?

3. Page 6, line 137-138 and 145-146: Disk diffusion was used, but never mentioned any resulting findings in terms of correlations with MIC in the rest of the manuscript.

4. Page 8, Fig 1 (C) and (D) should be eliminated.

5. Page 9, Fig 3: please keep only *A. hydrophila* data. please eliminate (B).

6. Page 10, Fig 4: please keep only *A. hydrophila* data. Please eliminate (B).

7. Page 11, Fig 5: please keep only *A. hydrophila* data.

8. Please cite a reference regarding SHD or "susceptible at high doses" for *Aeromonas*.

9. Page 11-12: the hypothesis regarding the resistance due to hyperexpression of the listed class B, C, or D could benefit from references that really used genetic approaches that verified those genetic determinants and their gene expression variations. Additionally, treatment outcomes from this group of patients would greatly enhance the understanding of the occult presence of these gene determinants and their implication in treatment.

10. *A. hydrophila* has been misspelled at places as *A. hydrophyla*.

Summary of the manuscript: An in vitro MIC survey of 99 clinical isolates of *Aeromonas* spp from French Guiana area.

Major comments: This study is well-designed with agents that have of clinical treatment importance. The data visualization is well organized. However, with the isolates of *A. caviae* of n=10, *A. veronii* of n=6, and other *Aeromonas* spp of n=4, the percent of susceptibility is not supported. Authors may consider present the MIC distributions associated with the 79 *A. hydrophila* isolates only for this report, other species can remain as brief descriptive of mentions, but not for species-specific antibiograms.

Minor comments:

1. Page 2, line 56: CASFM should have a full spell before the abbreviation.
2. Page 5, line 111: ..."oral relays for some CASES..."?
3. Page 6, line 137-138 and 145-146: Disk diffusion was used, but never mentioned any resulting findings in terms of correlations with MIC in the rest of the manuscript.
4. Page 8, Fig 1 (C) and (D) should be eliminated.
5. Page 9, Fig 3: please keep only *A. hydrophila* data. please eliminate (B).
6. Page 10, Fig 4: please keep only *A. hydrophila* data. Please eliminate (B).
7. Page 11, Fig 5: please keep only *A. hydrophila* data.
8. Please cite a reference regarding SHD or "susceptible at high doses" for *Aeromonas*.
9. Page 11-12: the hypothesis regarding the resistance due to hyperexpression of the listed class B, C, or D could benefit from references that really used genetic approaches that verified those genetic determinants and their gene expression variations. Additionally, treatment outcomes from this group of patients would greatly enhance the understanding of the occult presence of these gene determinants and their implication in treatment.
10. *A. hydrophila* has been misspelled at places as *A. hydrophyla*.

Subject: Spectrum00350-25 Decision Letter

Re: Spectrum00350-25 (Susceptibility Profile of *Aeromonas* spp. From Clinical Strains Isolated in French Guiana)

Reviewer #1:

General major comment on manuscript:

The most important feedback for the authors is that there needs to be at least a subset of isolates (if not for all 99 strains) for which ALL 3 AST method data are available: MIC from broth microdilution (BMD), zone diameter from disk diffusion and Etest MIC. Then, the BMD can be used to generate MIC distributions and compare susceptibility profiles to different agents across species. Additionally, the performance of Etest and disk diffusion can be evaluated relative to the reference BMD method. If not for all antimicrobial agents, then at least for the most clinically important agents (cefepime, ceftaz, pip-tazo etc).

Below are comments about further experiments and/or edits the authors can make to improve the overall quality of the study. I believe there are many more opportunities for the authors to increase the clinical utility of this study.

1. Please add a table in the methods that lists all the clinical MIC and disk diffusion BREAKPOINTS that were used in the study for each antimicrobial agent to interpret susceptibility testing results.

Response: We have added a table in the materials and methods section summarizing the breakpoints used.

2. I would highly recommend using only two types of clinical breakpoints to interpret susceptibility results: the CLSI M45 guidelines and EUCAST. Those are the two most common standards followed by clinical laboratories around the world. It keeps things simple and broadly applicable. I think the various breakpoints from CA-SFM are confusing and not widely used outside of France.

In Table 1, make sure to include a column to specify if the MIC or disk diffusion breakpoints are from CLSI or EUCAST. It would be informative to have this table contrasting the CLSI

versus EUCAST breakpoints for the agents where they are available. If there are differences between CLSI and EUCAST breakpoints for the same agent, then perform analysis with both S/I/R or S/R/ATU criteria to compare how many strains would categorize as susceptible or non-susceptible.

Response: As you recommended, we have chosen EUCAST as the reference. This did not imply any modification in the interpretation of our initial results. Furthermore, we have specified this in the title of Table 2.

3. It can be difficult to compare antimicrobial susceptibility profiles across drugs because 3 different methods were used inconsistently. Some drugs were tested with disk diffusion (CAZ, FEP, ATM, SXT, CIP, LEV) while others were tested with Etest (CTX), some had both disk and Etest (CAZ, CTX, FEP) and TZP was tested with broth microdilution. Given that the goal of the study is to characterize the susceptibility profiles of these strains and compare susceptibility to different antimicrobials, it is important to use standard methods with appropriate quality control strains.

- Is it possible to test all antimicrobial agents (or the most important ones used clinically) with Broth Microdilution (BMD)? BMD is considered the gold standard reference method for susceptibility testing.

- Considering that this is a single center study with microbiological data (in the absence of clinical data), it is important to make sure the right methods were used for AST to be able to compare susceptibility across agents and species. If resources are limited, choose 25-30 representative isolates of *Aeromonas hydrophila* based on the MIC/ disk diffusion data already available (across a range of MICs or zone diameters, some more susceptible, some resistant) plus all 10 *A. caviae* strains, 6 *A. veronii* strains and the 4 other *Aeromonas* spp strains. Even if testing is performed on a smaller subset of strains (~40-50 strains) rather than all 99 strains, the conclusions drawn will be more robust with reliable data.

- Either with all 99 strains or with a smaller representative subset of strains, I think it is important to have data to compare all three AST methods for the most clinically important antimicrobials. BMD, Etest, and disk diffusion. If resources are limited, can prioritize

cefepime, piperacillin-tazobactam, ceftazidime, bactrim (SXT), aztreonam. If more supplies available, can pick a fluoroquinolone (cipro or levo).

 Use MICs by broth microdilution (BMD) to generate main antimicrobial susceptibility profiling data. Please include the following data points: range of MIC values, MIC50 value, MIC90 value, % susceptible / % intermediate / % resistant for each agent across both CLSI breakpoints and EUCAST (whichever are available). Consider adding a TABLE summarizing the AST data and breaking it down for all strains tested, then across species to compare.

 Evaluate performance of Etest relative to BMD. Compare MIC range and MIC50/MIC90 values between Etest versus BMD. Calculate % essential agreement, % categorical agreement (across both CLSI and EUCAST breakpoints for comparison), very major errors, major errors and minor errors. Can use CLSI M23 criteria for error rates to determine if performance of Etest is acceptable or not. Please consider creating a table summarizing the method evaluation data across antimicrobials and species.

 Evaluate performance of disk diffusion relative to BMD. Calculate % categorical agreement (across both CLSI and EUCAST breakpoints for comparison), very major errors, major errors and minor errors. Create a Scattergram figure to visually show disk-diffusion-to-MIC correlation where the X axis has zone diameters and the Y axis has MIC values. If needed, look at this paper for reference on what the scattergrams can look like for each drug: <https://journals.asm.org/doi/10.1128/jcm.01757-19> (pubmed PMID:31996445). Same as before, you can use CLSI M23 criteria to then determine if performance of disk diffusion is acceptable or not.

Response: The objective of this article is not to compare the methods with each other. We mainly want to have a general idea of the sensitivities of *Aeromonas* spp in French Guiana, particularly regarding piperacillin-tazobactam and cefotaxime, as these molecules are widely used in the therapeutic protocols for *Aeromonas* spp infections in our hospital. For some molecules, we used several methods solely to confirm sensitivity or resistance. For clarity, I have removed duplicate methods in the materials and methods section for the molecules concerned.

4. Were quality control strains set up with each testing run? It is important to ensure the potency of the antimicrobials is at expected levels. Consider setting up QC strains with defined MIC ranges for each agent for each day of experiments. For example, consider setting up disk diffusion with ATCC strains like E. coli 25922 (CLSI M100 guidelines have tables with MIC and disk diffusion QC ranges for specific ATCC strains) and others that might be needed for each antimicrobial. Read QC results alongside the test strains to make sure data is reliable and MICs aren't falsely low or high.

Response: In our routine operations, we regularly test ATCC25922 for our AST methods since 2018.

5. Figure 2 legend says these are global MIC distributions for TZP but I don't believe that is true? Modify accordingly. The word "global" generally means the strains were collected from different geographic regions around the world. Please include # of strains that the graphs are based on in each panel (A-D). For example, how many strains are the distributions in Figure 2A based on? If 99 then specify.

Response: We deleted the word "global" from the legend of figure 2, 3, and 4. We added the number of strains on which the test was performed in the legend of the figures.

6. What media manufacturers of Mueller Hinton Agar were used for disk diffusion and Etest? Please specify this in the methods section. There can be variability to AST results across different brands of Mueller Hinton Agar. If possible, you can choose a smaller representative subset of isolates (can be as little as 20 strains) to perform disk diffusion and Etest across two or three brands of Mueller Hinton agar. It can show you if these AST methods are precise and reproducible for Aeromonas.

Response: We add the manufacturers of Mueller Hinton Agar. We do not have other brands of Mueller Hinton Agar. However, the objective of this article is not to compare the methods each other but to gain an understanding of the epidemiology of resistance to Aeromonas spp.

7. While comparing performance of disk diffusion or Etest relative to BMD, if there are very major and major errors, consider performing repeat testing to determine if those errors persist or resolve.

Response: Our objective is not to compare Etest and BMD. We used Etest for Cefotaxim and Ceftazidim and BMD for piperacillin Tazobactam because these are the methods we use to determine the MICs of these molecules in our laboratory.

8. Consider modifying the MIC distribution graphs after final experiments and include bars for MIC values from BMD and distinct colored bars for MIC values from Etest for comparison. Visually, it would help to see if the Etest tends to undercall or overcall resistance compared to BMD.

Response: Our work is an epidemiologic study and did not aim to compare the AST methods. For this, we believe that comparing the AST methods would be interesting but will answer another question rather than what we asked in the objectives of this study.

9. For all other drugs besides TZP like CTX, CAZ, FEP and ATM-- I think MIC values from broth microdilution are important for more accurate conclusions about the susceptibility profiles of different *Aeromonas* species. Since this is a big piece of manuscript, it seems important to make sure the data is robust. This applies to Figures 3, 4 and 5. Since there is Etest MIC data available also, the graphs can include bars for MICs from BMD versus different color bars for MICs from Etest for comparison.

10. Figure 5: A graph like this is drawing significant conclusions about *Aeromonas* susceptibility to different agents and would need to be based on broth microdilution-based MIC data. In this case, there are different methods used for different drugs and the clinical breakpoints are also different so it is very confusing.

Response : We do not have microdilution MICs for these molecules.

Reviewer #2:

Major comments: This study is well-designed with agents that have of clinical treatment importance. The data visualization is well organized. However, with the isolates of *A. caviae* of n=10, *A. veronii* of n=6, and other *Aeromonas* spp of n=4, the percent of susceptibility is not supported. Authors may consider present the MIC distributions associated with the 79 *A. hydrophila* isolates only for this report, other species can remain as brief descriptive of mentions, but not for species-specific antibiograms.

Minor comments:

1. Page 2, line 56: CASFM should have a full spell before the abbreviation. **According to a reviewer's recommendations, we have replaced CASFM with EUCAST.**
2. Page 5, line 111: ..."oral relays for some CASES..."? **we modified**
3. Page 6, line 137-138 and 145-146: Disk diffusion was used, but never mentioned any resulting findings in terms of correlations with MIC in the rest of the manuscript. **The objective of this article is not to compare the methods with each other. We mainly want to have a general idea of the sensitivities of *Aeromonas* spp in French Guiana, particularly regarding piperacillin-tazobactam and cefotaxime, as these molecules are widely used in the therapeutic protocols for *Aeromonas* spp infections in our hospital. For some molecules, we used several methods solely to confirm sensitivity or resistance. For clarity, I have removed duplicate methods in the materials and methods section for the molecules concerned.**
4. Page 8, Fig 1 (C) and (D) should be eliminated. **We modified**
5. Page 9, Fig 3: please keep only *A. hydrophila* data. please eliminate (B). **We modified**
6. Page 10, Fig 4: please keep only *A. hydrophila* data. Please eliminate (B). **We modified**
7. Page 11, Fig 5: please keep only *A. hydrophila* data. **Agreed, I have added a specific graph dedicated to *A. hydrophila*.**
8. Please cite a reference regarding SHD or "susceptible at high doses" for *Aeromonas*. **It is SHD according to EUCAST breackpoint. We added it.**
9. Page 11-12: the hypothesis regarding the resistance due to hyperexpression of the listed class B, C, or D could benefit from references that really used genetic approaches that verified those genetic determinants and their gene expression variations. Additionally, treatment outcomes from this group of patients would greatly enhance the understanding of the occult

presence of these gene determinants and their implication in treatment. Since 2018, we have observed this appearance of resistance during treatment on three occasions. We plan to write a case report on these cases, including, if possible, a genetic analysis of the strains to determine their phylogenetic relationship and the resistance mechanism involved.

10. A hydrophila has been misspelled at places as A hydrophyla. We modified

Re: Spectrum00350-25R1 (Susceptibility Profile of Aeromonas spp. From Clinical Strains Isolated in French Guiana)

Dear Dr. Vincent SAINTE-ROSE:

Thank you for the privilege of reviewing your work. Below you will find my comments, instructions from the Spectrum editorial office, and the reviewer comments.

Revision Guidelines

Sincerely,
Florence Doucet-Populaire
Editor
Microbiology Spectrum

Reviewer #1 (Comments for the Author):

I was pleased to review the revised manuscript. I commend the authors for making some improvements that added clarity to the manuscript. I urge the authors to consider improving their study design to ensure their conclusions have validity.

While the authors repeatedly state that the goal of the study was not to compare performance of methods, which is

understandable, they did emphasize that their main and only goal was to evaluate susceptibility profiles of Aeromonas in the region to clinically relevant antimicrobials. To this end, I think it is MORE important for the authors to use gold standard reference methods to determine susceptibility. The study uses different methodologies for different agents which makes it difficult to make comparisons across agents. It is critical to use 1 method across all antimicrobials in order for any comparison to be made. Or else, it is uncertain if any differences in susceptibility between agent #1 and agent #2 are due to real differences or due to the fact that different susceptibility testing methods were used. The study uses agar diffusion for some agents, disk diffusion for some and gradient strips for others but attempts to make comparisons across all agents which is a key methodological flaw in study design. Since the susceptibility patterns are the only broad conclusion of the entire study, I'm concerned that not using robust study design will affect validity of the conclusions drawn. I urge the authors to use a reference gold standard method and 1 method across all agents.

Subject: Spectrum00350-25 Decision Letter

Re: Spectrum00350-25 (Susceptibility Profile of *Aeromonas* spp. From Clinical Strains Isolated in French Guiana)

Reviewer #1:

General major comment on manuscript:

The most important feedback for the authors is that there needs to be at least a subset of isolates (if not for all 99 strains) for which ALL 3 AST method data are available: MIC from broth microdilution (BMD), zone diameter from disk diffusion and Etest MIC. Then, the BMD can be used to generate MIC distributions and compare susceptibility profiles to different agents across species. Additionally, the performance of Etest and disk diffusion can be evaluated relative to the reference BMD method. If not for all antimicrobial agents, then at least for the most clinically important agents (cefepime, ceftaz, pip-tazo etc).

Below are comments about further experiments and/or edits the authors can make to improve the overall quality of the study. I believe there are many more opportunities for the authors to increase the clinical utility of this study.

1. Please add a table in the methods that lists all the clinical MIC and disk diffusion BREAKPOINTS that were used in the study for each antimicrobial agent to interpret susceptibility testing results.

Response: We have added a table in the materials and methods section summarizing the breakpoints used.

2. I would highly recommend using only two types of clinical breakpoints to interpret susceptibility results: the CLSI M45 guidelines and EUCAST. Those are the two most common standards followed by clinical laboratories around the world. It keeps things simple and broadly applicable. I think the various breakpoints from CA-SFM are confusing and not widely used outside of France.

In Table 1, make sure to include a column to specify if the MIC or disk diffusion breakpoints are from CLSI or EUCAST. It would be informative to have this table contrasting the CLSI

versus EUCAST breakpoints for the agents where they are available. If there are differences between CLSI and EUCAST breakpoints for the same agent, then perform analysis with both S/I/R or S/R/ATU criteria to compare how many strains would categorize as susceptible or non-susceptible.

Response: As you recommended, we have chosen EUCAST as the reference. This did not imply any modification in the interpretation of our initial results. Furthermore, we have specified this in the title of Table 2.

3. It can be difficult to compare antimicrobial susceptibility profiles across drugs because 3 different methods were used inconsistently. Some drugs were tested with disk diffusion (CAZ, FEP, ATM, SXT, CIP, LEV) while others were tested with Etest (CTX), some had both disk and Etest (CAZ, CTX, FEP) and TZP was tested with broth microdilution. Given that the goal of the study is to characterize the susceptibility profiles of these strains and compare susceptibility to different antimicrobials, it is important to use standard methods with appropriate quality control strains.

- Is it possible to test all antimicrobial agents (or the most important ones used clinically) with Broth Microdilution (BMD)? BMD is considered the gold standard reference method for susceptibility testing.

- Considering that this is a single center study with microbiological data (in the absence of clinical data), it is important to make sure the right methods were used for AST to be able to compare susceptibility across agents and species. If resources are limited, choose 25-30 representative isolates of *Aeromonas hydrophila* based on the MIC/ disk diffusion data already available (across a range of MICs or zone diameters, some more susceptible, some resistant) plus all 10 *A. caviae* strains, 6 *A. veronii* strains and the 4 other *Aeromonas* spp strains. Even if testing is performed on a smaller subset of strains (~40-50 strains) rather than all 99 strains, the conclusions drawn will be more robust with reliable data.

- Either with all 99 strains or with a smaller representative subset of strains, I think it is important to have data to compare all three AST methods for the most clinically important antimicrobials. BMD, Etest, and disk diffusion. If resources are limited, can prioritize

cefepime, piperacillin-tazobactam, ceftazidime, bactrim (SXT), aztreonam. If more supplies available, can pick a fluoroquinolone (cipro or levo).

 Use MICs by broth microdilution (BMD) to generate main antimicrobial susceptibility profiling data. Please include the following data points: range of MIC values, MIC50 value, MIC90 value, % susceptible / % intermediate / % resistant for each agent across both CLSI breakpoints and EUCAST (whichever are available). Consider adding a TABLE summarizing the AST data and breaking it down for all strains tested, then across species to compare.

 Evaluate performance of Etest relative to BMD. Compare MIC range and MIC50/MIC90 values between Etest versus BMD. Calculate % essential agreement, % categorical agreement (across both CLSI and EUCAST breakpoints for comparison), very major errors, major errors and minor errors. Can use CLSI M23 criteria for error rates to determine if performance of Etest is acceptable or not. Please consider creating a table summarizing the method evaluation data across antimicrobials and species.

 Evaluate performance of disk diffusion relative to BMD. Calculate % categorical agreement (across both CLSI and EUCAST breakpoints for comparison), very major errors, major errors and minor errors. Create a Scattergram figure to visually show disk-diffusion-to-MIC correlation where the X axis has zone diameters and the Y axis has MIC values. If needed, look at this paper for reference on what the scattergrams can look like for each drug: <https://journals.asm.org/doi/10.1128/jcm.01757-19> (pubmed PMID:31996445). Same as before, you can use CLSI M23 criteria to then determine if performance of disk diffusion is acceptable or not.

Response: The objective of this article is not to compare the methods with each other. We mainly want to have a general idea of the sensitivities of *Aeromonas* spp in French Guiana, particularly regarding piperacillin-tazobactam and cefotaxime, as these molecules are widely used in the therapeutic protocols for *Aeromonas* spp infections in our hospital. For some molecules, we used several methods solely to confirm sensitivity or resistance. For clarity, I have removed duplicate methods in the materials and methods section for the molecules concerned.

4. Were quality control strains set up with each testing run? It is important to ensure the potency of the antimicrobials is at expected levels. Consider setting up QC strains with defined MIC ranges for each agent for each day of experiments. For example, consider setting up disk diffusion with ATCC strains like E. coli 25922 (CLSI M100 guidelines have tables with MIC and disk diffusion QC ranges for specific ATCC strains) and others that might be needed for each antimicrobial. Read QC results alongside the test strains to make sure data is reliable and MICs aren't falsely low or high.

Response: In our routine operations, we regularly test ATCC25922 for our AST methods since 2018.

5. Figure 2 legend says these are global MIC distributions for TZP but I don't believe that is true? Modify accordingly. The word "global" generally means the strains were collected from different geographic regions around the world. Please include # of strains that the graphs are based on in each panel (A-D). For example, how many strains are the distributions in Figure 2A based on? If 99 then specify.

Response: We deleted the word "global" from the legend of figure 2, 3, and 4. We added the number of strains on which the test was performed in the legend of the figures.

6. What media manufacturers of Mueller Hinton Agar were used for disk diffusion and Etest? Please specify this in the methods section. There can be variability to AST results across different brands of Mueller Hinton Agar. If possible, you can choose a smaller representative subset of isolates (can be as little as 20 strains) to perform disk diffusion and Etest across two or three brands of Mueller Hinton agar. It can show you if these AST methods are precise and reproducible for Aeromonas.

Response: We add the manufacturers of Mueller Hinton Agar. We do not have other brands of Mueller Hinton Agar. However, the objective of this article is not to compare the methods each other but to gain an understanding of the epidemiology of resistance to Aeromonas spp.

7. While comparing performance of disk diffusion or Etest relative to BMD, if there are very major and major errors, consider performing repeat testing to determine if those errors persist or resolve.

Response: Our objective is not to compare Etest and BMD. We used Etest for Cefotaxim and Ceftazidim and BMD for piperacillin Tazobactam because these are the methods we use to determine the MICs of these molecules in our laboratory.

8. Consider modifying the MIC distribution graphs after final experiments and include bars for MIC values from BMD and distinct colored bars for MIC values from Etest for comparison. Visually, it would help to see if the Etest tends to undercall or overcall resistance compared to BMD.

Response: Our work is an epidemiologic study and did not aim to compare the AST methods. For this, we believe that comparing the AST methods would be interesting but will answer another question rather than what we asked in the objectives of this study.

9. For all other drugs besides TZP like CTX, CAZ, FEP and ATM-- I think MIC values from broth microdilution are important for more accurate conclusions about the susceptibility profiles of different *Aeromonas* species. Since this is a big piece of manuscript, it seems important to make sure the data is robust. This applies to Figures 3, 4 and 5. Since there is Etest MIC data available also, the graphs can include bars for MICs from BMD versus different color bars for MICs from Etest for comparison.

10. Figure 5: A graph like this is drawing significant conclusions about *Aeromonas* susceptibility to different agents and would need to be based on broth microdilution-based MIC data. In this case, there are different methods used for different drugs and the clinical breakpoints are also different so it is very confusing.

Response : We do not have microdilution MICs for these molecules.

Reviewer #2:

Major comments: This study is well-designed with agents that have of clinical treatment importance. The data visualization is well organized. However, with the isolates of *A. caviae* of n=10, *A. veronii* of n=6, and other *Aeromonas* spp of n=4, the percent of susceptibility is not supported. Authors may consider present the MIC distributions associated with the 79 *A. hydrophila* isolates only for this report, other species can remain as brief descriptive of mentions, but not for species-specific antibiograms.

Minor comments:

1. Page 2, line 56: CASFM should have a full spell before the abbreviation. **According to a reviewer's recommendations, we have replaced CASFM with EUCAST.**
2. Page 5, line 111: ..."oral relays for some CASES..."? **we modified**
3. Page 6, line 137-138 and 145-146: Disk diffusion was used, but never mentioned any resulting findings in terms of correlations with MIC in the rest of the manuscript. **The objective of this article is not to compare the methods with each other. We mainly want to have a general idea of the sensitivities of *Aeromonas* spp in French Guiana, particularly regarding piperacillin-tazobactam and cefotaxime, as these molecules are widely used in the therapeutic protocols for *Aeromonas* spp infections in our hospital. For some molecules, we used several methods solely to confirm sensitivity or resistance. For clarity, I have removed duplicate methods in the materials and methods section for the molecules concerned.**
4. Page 8, Fig 1 (C) and (D) should be eliminated. **We modified**
5. Page 9, Fig 3: please keep only *A. hydrophila* data. please eliminate (B). **We modified**
6. Page 10, Fig 4: please keep only *A. hydrophila* data. Please eliminate (B). **We modified**
7. Page 11, Fig 5: please keep only *A. hydrophila* data. **Agreed, I have added a specific graph dedicated to *A. hydrophila*.**
8. Please cite a reference regarding SHD or "susceptible at high doses" for *Aeromonas*. **It is SHD according to EUCAST breackpoint. We added it.**
9. Page 11-12: the hypothesis regarding the resistance due to hyperexpression of the listed class B, C, or D could benefit from references that really used genetic approaches that verified those genetic determinants and their gene expression variations. Additionally, treatment outcomes from this group of patients would greatly enhance the understanding of the occult

presence of these gene determinants and their implication in treatment. Since 2018, we have observed this appearance of resistance during treatment on three occasions. We plan to write a case report on these cases, including, if possible, a genetic analysis of the strains to determine their phylogenetic relationship and the resistance mechanism involved.

10. A hydrophila has been misspelled at places as A hydrophyla. We modified

Reviewer #1 (Comments for the Author):

I was pleased to review the revised manuscript. I commend the authors for making some improvements that added clarity to the manuscript. I urge the authors to consider improving their study design to ensure their conclusions have validity.

While the authors repeatedly state that the goal of the study was not to compare performance of methods, which is understandable, they did emphasize that their main and only goal was to evaluate susceptibility profiles of *Aeromonas* in the region to clinically relevant antimicrobials. To this end, I think it is MORE important for the authors to use gold standard reference methods to determine susceptibility. The study uses different methodologies for different agents which makes it difficult to make comparisons across agents. It is critical to use 1 method across all antimicrobials in order for any comparison to be made. Or else, it is uncertain if any differences in susceptibility between agent #1 and agent #2 are due to real differences or due to the fact that different susceptibility testing methods were used. The study uses agar diffusion for some agents, disk diffusion for some and gradient strips for others but attempts to make comparisons across all agents which is a key methodological flaw in study design. Since the susceptibility patterns are the only broad conclusion of the entire study, I'm concerned that not using robust study design will affect validity of the conclusions drawn. I urge the authors to use a reference gold standard method and 1 method across all agents.

Reponse :

We thank the reviewer for this pertinent comment. As we are currently unable to perform reference susceptibility testing for all antibiotics, we have decided to temper our discussion by adding the following sentence : « Furthermore it is important to note that in our retrospective study, several methods were used to determine antibiotic susceptibility, including disk diffusion, E-test strips, and microdilution strips. As disk diffusion is not the reference method, further studies are needed to confirm our results regarding the following antibiotics: ATM, FEP, CIP, LEV, and SXT. »

Re: Spectrum00350-25R2 (Susceptibility Profile of Aeromonas spp. From Clinical Strains Isolated in French Guiana)

Dear Dr. Vincent SAINTE-ROSE:

Your manuscript has been accepted, and I am forwarding it to the ASM production staff for publication. Your paper will first be checked to make sure all elements meet the technical requirements. ASM staff will contact you if anything needs to be revised before copyediting and production can begin. Otherwise, you will be notified when your proofs are ready to be viewed.

Sincerely,
Florence Doucet-Populaire
Editor
Microbiology Spectrum